# Magnetic-Field-Induced Improvement of Photothermal Sterilization Performance by Fe_3_O_4_@SiO_2_@Au/PDA Nanochains

**DOI:** 10.3390/ma16010387

**Published:** 2022-12-31

**Authors:** Kezhu Xu, Qunling Fang, Jing Wang, Ailing Hui, Shouhu Xuan

**Affiliations:** 1School of Food and Biological Engineering, Hefei University of Technology, Hefei 230009, China; 2CAS Key Laboratory of Mechanical Behavior and Design of Materials, Department of Modern Mechanics, University of Science and Technology of China, Hefei 230027, China

**Keywords:** antibacterial, photothermal effect, magnetic, nanochain, sterilization

## Abstract

Due to the abuse of antibiotics, the sensitivity of patients to antibiotics is gradually reduced. This work develops a Fe_3_O_4_@SiO_2_@Au/PDA nanochain which shows an interesting magnetic-field-induced improvement of its photothermal antibacterial property. First, SiO_2_ was wrapped on Fe_3_O_4_ nanospheres assembled in a chain to form a Fe_3_O_4_@SiO_2_ nanocomposite with a chain-like nanostructure. Then, the magnetic Fe_3_O_4_@SiO_2_@Au/PDA nanochains were prepared using in situ redox-oxidization polymerization. Under the irradiation of an 808 nm NIR laser, the temperature rise of the Fe_3_O_4_@SiO_2_@Au/PDA nanochain dispersion was obvious, indicating that they possessed a good photothermal effect. Originating from the Fe_3_O_4_, the Fe_3_O_4_@SiO_2_@Au/PDA nanochain showed a typical soft magnetic behavior. Both the NIR and magnetic field affected the antimicrobial performance of the Fe_3_O_4_@SiO_2_@Au/PDA nanochains. *Escherichia coli* and *Staphylococcus aureus* were used as models to verify the antibacterial properties. The experimental results showed that the Fe_3_O_4_@SiO_2_@Au/PDA nanochains exhibited good antibacterial properties under photothermal conditions. After applying a magnetic field, the bactericidal effect was further significantly enhanced. The above results show that the material has a broad application prospect in inhibiting the growth of bacteria.

## 1. Introduction

There are a large number of pathogenic bacteria in the environment of daily life, which seriously threaten people’s physical and mental health. In order to meet the increasingly sophisticated needs of the public for health and safety, scientists have been trying to develop a variety of efficient and safe antibacterial drugs. So far, the use of antibiotics is still the most widespread method to treat diseases caused by bacterial infections [1,2,3]. As bacterial infections become more and more serious, antibiotics are used in large quantities, which leads to a great increase in the resistance of bacteria, thereby reducing the therapeutic effect of antibiotics. Especially in the case of local treatment, such as skin lesion or breast cancers, safer treatment is urgently required for the application to have precise targeting, avoiding greater risks to the patient. Therefore, researchers are actively looking for effective methods to treat bacterial infections without causing high drug resistance.

The photothermal therapy of bacteria under near-infrared light irradiation has attracted wide attention due to its unique treating effects [4,5,6]. Photothermal therapy (PTT), as a new technology for the treatment of bacterial infections, has many advantages such as low damage to the body, good antibacterial effects, and low toxicity and side effects. Since photothermal therapy physically kills bacteria through thermal ablation, it can effectively reduce the generation of bacterial drug resistance, thus showing broad application prospects [7,8,9,10]. Considering that PTT also shows some drawbacks such as non-specificity and low penetration, intensive research has been conducted to improve PTT’s performance and clarify its mechanism. With the development of nanotechnology, the combination of photothermal therapy with nanomaterials has introduced novel possibilities for antibacterial treatment. During the past decade, various nanomaterials with unique photothermal activity, such as noble metals, semiconductor nanocrystals, and carbon materials, have been intensively studied [11,12,13,14,15]. In addition to their high photothermal effects, multifunctional PTT nanoplatforms with fewer side effects are of special vital importance in developing novel therapeutics against bacterial infections. As a result, core/shell-structured PTT nanocomposites have become attractive since their high surface-modification activity, high biocompatibility, high light-to-heat conversion efficiency, and low toxicity can effectively improve antibacterial performance [16,17].

Magnetic nanocomposites could introduce photothermal nanoplatforms with advantages such as high photothermal conversion, strong magnetic properties, and easy separation [18]. Fe_3_O_4_ nanoparticles have been explored in different bacterial and drug-delivery nanosystems due to their wonderful magnetic characteristics. Based on magnetic Fe_3_O_4_ nanocrystals, many photothermal nanoplatforms have been studied. Li et al. designed a polydopamine-modified Fe_3_O_4_ nanocomposite as a magnetic therapeutic nanoagent for MRI-guided photothermal therapy (PTT) to treat cancer [19]. Zhang et al. successfully prepared Fe_3_O_4_@CuS photothermal microspheres using a simple chemical deposition method [20]. Getiren et al. synthesized a core-shell structured superparamagnetic Fe_3_O_4_@PPy nanocomposite with a strong near-infrared absorption [21]. These materials all have a spherical structure and exhibit excellent performance in antibacterial applications. Recently, it was found that magnetic NiFe_2_O_4_@Au/PDA nanospheres exhibit a high antibacterial performance [22]. Under an applied magnetic field, the photothermal antibacterial effect increased due to the magnetically induced forces on the magnetic nanospheres. Moreover, this wonderful magnetolytic effect has also been shown in the MXene@Fe_3_O_4_/Au/PDA nanosheet photothermal antibacterial system [23]. Obviously, the nanostructure of the magnetic nanocomposites also plays an important role in magneto-photothermal antibacterial performance. As a result, more work should be conducted to develop special magnetic nanostructures and investigate the magnetic-field-induced improvement of photothermal sterilization performance.

Polydopamine (PDA) is a wonderful biocompatible polymer which has the advantages of simple preparation, high adhesion, and easy functionalization. Due to its low cytotoxicity and good binding ability to target cells, PDA has been widely applied to cover nanomaterials to form various functional nanomedicines [22,23]. In this work, a NIR-light-activated antibacterial system based on Fe_3_O_4_@SiO_2_@Au/PDA nanochains is developed to realize the coupling of high magnetolytic and photothermal antibacterial activity and low cytotoxicity. Due to the presence of a polydopamine (PDA) shell, sandwiched Au nanocrystals, and an inner magnetic Fe_3_O_4_ core, the biocompatible Fe_3_O_4_@SiO_2_@Au/PDA nanochains have both photothermal effects and magnetic-field-guided effects. When the concentration of Fe_3_O_4_@SiO_2_@Au/PDA nanochains is 500 μg/mL, the temperature rises to 57 °C if irradiated by an 808 nm laser for 5 min, and the photothermal conversion efficiency is 44.26%. Moreover, the coupled photothermal therapy and magnetic-guided antibacterial properties against *Escherichia coli* and *Staphylococcus aureus* were clearly observed under the simultaneous introduction of NIR and a magnetic field. This material shows excellent magnetolytic–photothermal coupling inhibitory effects, and thus possesses a high potential for future antibacterial strategy development.

## 2. Experimental Section

### 2.1. Materials

Iron(III) chloride hexahydrate (FeCl_3_·6H_2_O), ethylene glycol, diethylene glycol, sodium acetate anhydrous, polyacrylic acid (PAA), ammonia monohydrate (NH_3_·H_2_O), tetraethyl orthosilicate (TEOS), ethanol (EtOH), gold(III) chloride (HAuCl_4_), trihydroxymethyl aminomethane (Tris), hydrochloric acid, sodium citrate (C6H_5_O_7_·2H_2_O), (3-aminopropyl)-triethoxysilane (APTES, C_9_H_23_NO_3_Si), and 3-hydroxytyrosine hydrochloride (DA-HCl) were bought from Aladdin Reagent (Shanghai, China). All chemicals were used as received without any further purification. The water used in the experiment was ultrapure water.

### 2.2. Preparation of Fe_3_O_4_@SiO_2_ Nanochain

First, Fe_3_O_4_ nanoparticles were synthesized through a one-pot solvothermal reaction [24]. Generally, FeCl_3_·6H_2_O (1.08 g) was dissolved in a mixed solution of ethylene glycol (10 mL) and diethylene glycol (30 mL). After being completely dissolved, polyacrylamide (PAA) was added under continuous stirring. After thorough mixing, the homogeneous mixture was transferred and sealed in a stainless steel autoclave lined with polytetrafluoroethylene (PTFE) and heated to 200 °C. Twelve hours later, the system was cooled to room temperature and the obtained Fe_3_O_4_ nanoparticles were washed alternately with deionized water and ethanol 4 times. Then, the product was dried using a static magnetic field under vacuum at 40 °C for 12 h.

Next, the Fe_3_O_4_@SiO_2_ was prepared. First, 10 mg Fe_3_O_4_ was dispersed in ethanol (30 mL) and deionized water (3 mL). After it was uniformly dispersed, 1.8 mL of ammonia monohydrate was added. After sonicating for 10 min, 0.2 mL of TEOS was added. The reaction was conducted for 1.5 h. After the reaction, the product was collected from the solution with a magnet. The obtained black solid was washed with deionized ethanol several times, and dried under vacuum for 12 h to obtain a dark gray powder.

### 2.3. Preparation of Core/Shell Structure Fe_3_O_4_@SiO_2_@Au/PDA Nanochain

The aforementioned Fe_3_O_4_@SiO_2_ was used as the core template to synthesize the Fe_3_O_4_@SiO_2_@Au/PDA core/shell nanochains. First, 15 mg Fe_3_O_4_@SiO_2_ was ultrasonically dispersed in 50 mL of C_2_H_5_OH. Next, 15 mg of sodium citrate and 60 μL of HAuCl_4_ were added to form a mixed solution. Then, 45 mg dopamine was added to 45 mL of Tris (pH = 8.5) to form another solution. After complete dissolution, the two solutions were mixed. After 3 h sonication, the product was collected from the solution using magnetic separation. The obtained black solid was washed alternately with deionized water and ethanol, and dried under vacuum for 12 h to obtain a black powder.

### 2.4. The Photothermal Effect of Fe_3_O_4_@SiO_2_@Au/PDA Nanochain

The prepared Fe_3_O_4_@SiO_2_@Au/PDA standard solution was diluted into 100 μg/mL, 200 μg/mL, 300 μg/mL, 400 μg/mL, and 500 μg/mL solutions. A 2 mL volume of Fe_3_O_4_@SiO_2_@Au/PDA solution at a concentration of 100 μg/mL was placed in a 4 mL centrifuge tube. Under the irradiation of an 808 nm near-infrared laser, the temperature of the solution was recorded with an infrared camera every 20 s. The influence of the different concentrations of Fe_3_O_4_@SiO_2_@Au/PDA solutions on the temperature was also recorded using this method.

### 2.5. Photothermal Therapy (PTT) and Magnetic Guidance Effect Synergistic Sterilization of the Fe_3_O_4_@SiO_2_@Au/PDA Nanochain

First, a certain amount of LB agar medium and broth medium were prepared in proportion. After high-temperature sterilization and cooling, the LB agar medium was shaken in the ultra-clean platform and then was poured into Petri dishes. The Petri dishes were placed horizontally. Then, a certain amount of the original bacterial solution was added to the sterilized broth medium for 10 h to obtain the bacteria in the logarithmic growth phase. Samples with different concentrations were prepared with sterilized LB agar medium, and the experimental group and blank control were set. *Escherichia coli* or *Staphylococcus aureus* were added to the experimental groups and treated under different conditions (Group 1: Petri dishes were irradiated with a laser only; Group 2: after laser irradiation, magnetic field treatment was performed). Bacterial suspensions were treated with an NIR laser (L, 2.5 W cm^−2^) and/or a rotating magnetic field (RMF, 15 mT, 350 rpm), and the three groups were labeled as Control, PTT, and PTT + Magnet, respectively. Except for the control group, the treatment time was 6 min, where PTT + Magnet was the combined action of NIR laser and rotating magnetic field for 6 min. During the magnetic treatment, the magnetic field was applied using a magnetic stirrer directly under the solution. In the control group, only bacteria and materials were added without any treatment. The above solution was placed in a 37 °C constant-temperature incubator for 24 h. The co-culture bacterial liquid was gradually diluted with a 24-well plate as a container, and a certain amount of the diluted bacterial liquid was absorbed and coated on the prepared Petri dish, and then placed in an incubator at 37 °C for cultivation for 24 h to observe the growth of the bacterial colony.

### 2.6. MTT Cell Assays of the Fe_3_O_4_@SiO_2_@Au/PDA Nanochains

The biocompatibility and cytotoxicity of Fe_3_O_4_@SiO_2_@Au/PDA nanochains were evaluated with MTT cell assays in human umbilical vein endothelial cells (HUVEC). Under standard conditions (37 °C, 5% CO_2_), 10% fetal bovine serum (FBS) was added to DMEM medium (Dulbecco’s Modified Eagle Medium, a medium containing various amino acids and glucose). HUVEC in the logarithmic growth phase were inoculated on 96-well cell culture plates (1 × 10^4^ cells per well, 200 μL) and incubated for 24 h. Then, fresh DMEM medium containing Fe_3_O_4_@SiO_2_@Au/PDA nanochains (with different gradient concentrations) was replaced and incubated for another 24 h. HUVEC were washed with fresh medium and cell viability was confirmed using a standard MTT assay according to the manufacturer’s protocol.

### 2.7. Characterization

The morphologies of the nanocomposites were studied using a JEM-2100 F field emission transmission electron microscope (FE-TEM, JEOL, Tokyo, Japan) operating at 200 kV acceleration voltage. The EDS spectra were investigated with the JEM-2100 F field emission transmission electron microscope (FE-TEM). Field emission scanning electron microscopy (FE-SEM) images were operated on a Regulus 8230 HR-FESEM (Hitachi, Tokyo, Japan). The XRD patterns were obtained with a Bruker D8 Advance diffractometer (Karlsruhe, Germany), which was equipped with graphite monochromatized Cu Kα radiation (λ =1.5406 Å). The X-ray photoelectron spectra (XPS) were recorded with an ESCALAB 250(Thermo Scientific, Walthamm, MA, USA). Thermogravimetric (TG) analysis was performed on a DTG-60H thermogravimetric instrument and the nanocomposites were analyzed under air flowing from room temperature to 800 °C at a heating rate of 10 °C min^−1^. Hysteresis loops were measured on a vibrating sample magnetometer (SQUID-VSM, QuantumDesign, San Diego, CA, USA) at 300 K.

## 3. Results and Discussion

Figure 1 shows the synthetic illustration of the Fe_3_O_4_@SiO_2_@Au/PDA nanochains. First, the Fe_3_O_4_ nanospheres were assembled to form chain-like nanostructures during synthesis. Then, the SiO_2_ was coated on the Fe_3_O_4_ nanochains to form the Fe_3_O_4_@SiO_2_ core/shell nanochains. At last, the Au/PDA hybrid shell further covered the above particles via an in situ redox-oxidization polymerization method through the reaction between HAuCl_4_ and dopamine [25]. Figure 1a shows the SEM of the obtained Fe_3_O_4_@SiO_2_@Au/PDA nanochains. It can be seen that the synthesized chain-like nanostructures have different lengths and their shapes are similar to candied haws. Figure 1b shows the TEM image of the product, which agrees well with the SEM analysis. The samples in the entire field of view are all chain-like structures (Figure 1c), indicating the successful synthesis of chain-like Fe_3_O_4_@SiO_2_@Au/PDA nanostrings. Figure 1d is the high-power transmission electron micrograph. It can be observed that the obtained Fe_3_O_4_@SiO_2_@Au/PDA nanochain has an obvious core/shell nanostructure. The outermost shell layer is a transparent PDA layer and it is uniformly coated on the surface of the nanochain. The sandwiched black layer is a discontinuous layer of Au nanoparticles and the inner black spherical core/shell structure is Fe_3_O_4_@SiO_2_ nanochain.

To further understand the inner nanostructure and composition of the final product, high-angle annular dark-field scanning electron microscopy (HAADF-STEM) and energy dispersive X-ray spectroscopy (EDX) elemental maps were created. As shown in Figure 2, the HAADF-STEM also confirms that the Fe_3_O_4_@SiO_2_@Au/PDA nanocomposite is a quaternary component core/shell nanochain (Figure 2a,b). By observing the element distribution map of Fe, it can be seen that the image size of the Fe element is small (about 70 nm) and it is mainly located in the core (Figure 2a,c). Clearly, the Fe elements collected with each other to form a chain-like nanostructure, which also demonstrates that the Fe_3_O_4_ nanochains are formed by the Fe_3_O_4_ nanospheres. Figure 2d illustrates that the size of Si is obviously larger than that of Fe, which indicates that SiO_2_ covers the surface of Fe_3_O_4_. After the SiO_2_ coating, the diameter of the nanochain increases to about 120 nm. The O element shows a similar distribution size with the Si, demonstrating that this immediate shell is composed of SiO_2_ (Figure 2e). Here, the Au/PDA layer is simultaneously coated on the surface of the Fe_3_O_4_@SiO_2_, thus the distribution size of Au is beyond the Si (Figure 2f). However, the distribution density of Au in the core is low and the Au nanocrystals are immobilized around the SiO_2_ with a non-continuous nature. This result matches the above TEM analysis. The diameters of the elemental profiles of C and N are approximately the same (Figure 2g,h), further proving that the PDA shell uniformly coats the surface of Fe_3_O_4_@SiO_2_. The size of the N is a little larger than the Au, demonstrating the Au nanocrystals are encapsulated by the PDA shell. Based on the above analysis, it can be concluded that the Fe_3_O_4_@SiO_2_@Au/PDA nanochain with core/shell structure was successfully synthesized.

The TEM was also applied to analyze the synthesis of the Fe_3_O_4_@SiO_2_@Au/PDA nanochains. Here, the Fe_3_O_4_ nanochains assembled by Fe_3_O_4_ nanospheres were firstly obtained using the solvothermal method (Figure 3a). During the preparation, they were dried under a static magnetic field in vacuum at 40 °C for 12 h; thus, the Fe_3_O_4_ nanospheres assembled to form a chain-like nanostructure since they tend to align with the magnetic flux lines. Due to the interaction of PAA residues in the Fe_3_O_4_ nanospheres, the chain-like nanostructure assembled from the Fe_3_O_4_ nanospheres is relatively stable (Figure 3b). Therefore, they can be applied as the chain template for the following coating process. In the preparation of Fe_3_O_4_@SiO_2_ nanochains, the chain-like Fe_3_O_4_ (a single Fe_3_O_4_ with 70 nm) was added to the ethanol solution and ultrasonically dispersed. After introducing ammonia and tetra-orthosilicate, the reaction was started and the Fe_3_O_4_@SiO_2_ nanochains were obtained (Figure 3c). The introduction of the SiO_2_ coating was to make the chain structure more stable. Obviously, the size of the nanochain increases and the Fe_3_O_4_@SiO_2_ nanochains show the typical core/shell nanostructure. Then, using the Fe_3_O_4_@SiO_2_ nanochain as the template, the Fe_3_O_4_@SiO_2_@Au/PDA could be successfully obtained with the reaction between HAuCl_4_ and dopamine. HAuCl_4_ quickly reacted with the dopamine monomer to form Au nanoparticles and PDA polymer chains. The in situ-formed Au nanocrystals were protected by the polydopamine (PDA) polymer and they tended to attach onto the surface of the Fe_3_O_4_@SiO_2_ nanochains to form Fe_3_O_4_@SiO_2_@Au/PDA nanochains (Figure 3d). Due to the low concentration of HAuCl_4_ in our synthesis, the Au nanocrystals showed a non-continuous layer on the Fe_3_O_4_@SiO_2_ and they were encapsulated by the external PDA shell.

The chainlike Fe_3_O_4_ nanostructure plays a critical role in forming the Fe_3_O_4_@SiO_2_@Au/PDA nanochains. As a comparison, a Fe_3_O_4_@SiO_2_@Au/PDA nanosphere would be obtained using the monodispersed Fe_3_O_4_ nanosphere as the starting material. Figure 4a indicates that the monodispersed Fe_3_O_4_ nanosphere has a diameter of about 70 nm. After the SiO_2_ coating using the sol-gel method, the diameter of the Fe_3_O_4_@SiO_2_ core/shell nanosphere increased to 130 nm, and the thickness of the SiO_2_ shell layer was about 30 nm (Figure 4b). After the further Au/PDA coating, the product showed an obvious four-layer structure, in which the outermost was the discontinuous Au layer and the continuous and uniform PDA layer (Figure 4c). Figure 4d shows a typical TEM image of a single Fe_3_O_4_@SiO_2_@Au/PDA nanosphere. It is observed that the diameter of Au is about 10 nm and the thickness of the PDA shell is about 20 nm, thus the average size of Fe_3_O_4_@SiO_2_@Au/PDA nanoparticles is about 200 nm.

In order to fully investigate the nanostructure of Fe_3_O_4_@SiO_2_@Au/PDA nanospheres, high-angle circular dark-field scanning TEM (HAADF-STEM) images were studied. As shown in Figure 5, TEM images of high-angle dark-field annular scanning indicated that Fe_3_O_4_@SiO_2_ core/shell nanostructure is present inside the Fe_3_O_4_@SiO_2_@Au/PDA nanocomposite. The Si element shows a weak hollow morphology, indicating the SiO_2_ covers the Fe_3_O_4_ nanosphere to form the core/shell nanostructure. In comparison to the Si and O elements, the distribution size of the Au is larger, which further proves the Au nanocrystals are immobilized on the surface of the SiO_2_ shell. The N element is detected in Fe_3_O_4_@SiO_2_@Au/PDA nanospheres, demonstrating that the whole nanosphere is encapsulated within a PDA shell. It can be found that below the Au/PDA hybrid layer is the SiO_2_ shell with a thickness of about 60 nm and the central core part is Fe_3_O_4_ superparamagnetic nanosphere. Based on the above analysis, it is concluded that the Fe_3_O_4_@SiO_2_@Au/PDA core/shell nanostructure was successfully prepared using the Fe_3_O_4_ nanosphere as the template. If the core changes to the Fe_3_O_4_ nanochain, the Fe_3_O_4_@SiO_2_@Au/PDA nanochain will be obtained.

The synthesis process of the sample can be tracked using thermogravimetry. As shown in Figure 6a. The weight loss rates of Fe_3_O_4_, Fe_3_O_4_@SiO_2_, and Fe_3_O_4_@SiO_2_@Au/PDA nanochains are 9.38 wt%, 9.8 wt%, and 17.1 wt%, respectively. With increasing temperature, the weight of Fe_3_O_4_ begins to decrease, which must originated from the decomposition of the organic groups and water contained in the Fe_3_O_4_ nanochains. When the surface of the Fe_3_O_4_ nanochain is coated with an SiO_2_ shell, its weight loss is about 4.2 wt% higher than that of Fe_3_O_4_ due to the high water absorbance in the SiO_2_ shell prepared with the sol-gel process. After further coating the Au/PDA layer on the Fe_3_O_4_@SiO_2_, the weight loss of Fe_3_O_4_@SiO_2_@Au/PDA nanocomposite is further increased. This result is owing to the PDA’s decomposition at high temperatures which causes a larger mass loss.

To further study the internal structure of the Fe_3_O_4_@SiO_2_@Au/PDA nanocomposite, XPS (X-ray photoelectron spectroscopy) was used to analyze the element composition on the surface of the samples. Figure 6b shows the XPS spectra of Fe_3_O_4_, Fe_3_O_4_@SiO_2_, and Fe_3_O_4_@SiO_2_@Au/PDA nanochains. The strong signal peaks of Fe, C, and O elements can be clearly observed in the XPS spectrum of Fe_3_O_4_. After coating with SiO_2_, an obvious signal peak of Si is detected in the Fe_3_O_4_@SiO_2_, which indicates the existence of the SiO_2_ shell. At the same time, it can be seen that the Fe signal peak disappears, which indicates that the SiO_2_ shell has successfully coated the Fe_3_O_4_ surface and blocked the Fe signal. When the Au/PDA layer covers the surface of the Fe_3_O_4_@SiO_2_ to form the Fe_3_O_4_@SiO_2_@Au/PDA nanochains, the typical Au signal peaks and N signal peaks can be observed. Here, the N element comes from the PDA layer. Based on the above analysis, it can be concluded that Fe_3_O_4_@SiO_2_@Au/PDA nanochains were successfully prepared using the simple method.

Figure 6c shows the XRD patterns of the Fe_3_O_4_, Fe_3_O_4_@SiO_2_, and Fe_3_O_4_@SiO_2_@Au/PDA nanochains. The characteristic peaks of all diffraction planes for Fe_3_O_4_ nanochains can be indexed as (220), (311), (440), (511), and (440) lattice planes with a face-centered cubic crystallization. Because the SiO_2_ is amorphous, no new diffraction peak is found in Fe_3_O_4_@SiO_2_. The XRD pattern of Fe_3_O_4_@SiO_2_ is similar to that of Fe_3_O_4_, which indicates that the SiO_2_ coating process does not affect the crystal phase of Fe_3_O_4_. When Au/PDA is further fixed on the Fe_3_O_4_@SiO_2_ surface, obvious Au diffraction peaks (111), (200), and (220) appeared at 38°, 44.1°, and 64.8°, respectively. This result also indicates the successful preparation of the Fe_3_O_4_@SiO_2_@Au/PDA nanochains. The magnetic properties of Fe_3_O_4_, Fe_3_O_4_@SiO_2_, and Fe_3_O_4_@SiO_2_/Au/PDA nanochains are shown in Figure 3d. Obviously, the magnetic properties of all samples originate from the Fe_3_O_4_ core. The saturation magnetization (*Ms*) of nanocomposites decreases with the addition of non-magnetic SiO_2_ and Au/PDA hybrid layers. Although the *Ms* value of Fe_3_O_4_@SiO_2_@Au/PDA (19.5 emu/g) is lower than those of Fe_3_O_4_@SiO_2_ (30.1 emu/g) and Fe_3_O_4_ (70.1 emu/g), its magnetic response capacity is sufficient to use a magnet to remove it from the reaction system. More importantly, the Fe_3_O_4_@SiO_2_@Au/PDA nanochains possess a soft magnetic characteristic (the coercivity of the nanoparticles was smaller than 50 Oe, as shown in the inset of Figure 6d), which must exhibit high potential application prospects in nanocatalysis and targeted drugs.

Due to the presence of Au and PDA, the Fe_3_O_4_@SiO_2_@Au/PDA nanochains show light absorption and photothermal conversion properties. As shown in Figure 7a, the Fe_3_O_4_@SiO_2_@Au/PDA nanochains have strong and wide absorbance in the ultraviolet to near-infrared range, which gives them great potential in the near infrared light (NIR)-triggered photothermal sterilization. With an increase in the laser irradiation time or Fe_3_O_4_@SiO_2_@Au/PDA concentration, the temperature of the nanochain suspension increases rapidly. After exposure for 300 s, the final temperature of the Fe_3_O_4_@SiO_2_@Au/PDA nanochain suspension with a concentration of 500 μg/mL was 57 °C. According to the temperature-rise curves of different samples in Figure 7b, the temperature increment of pure water is almost negligible under the same illumination time, while the temperature increment of Fe_3_O_4_@SiO_2_@Au/PDA nanochains is larger than the Fe_3_O_4_ and Fe_3_O_4_@SiO_2_ nanochains.

In addition, Fe_3_O_4_@SiO_2_@Au/PDA nanochains have excellent chemical stability and stable photothermal conversion properties. As shown in Figure 7c, the 500 μg/mL Fe_3_O_4_@SiO_2_@Au/PDA can still reach a similar temperature level after five bouts of NIR laser irradiation, which indicates that the Fe_3_O_4_@SiO_2_/Au/PDA nanochain has good photothermal stability and reusability. The photothermal conversion efficiency (η) is an important parameter to evaluate photothermal agents. According to the method established by Rope et al., [26] the accurate η value of the Fe_3_O_4_@SiO_2_@Au/PDA under 808 nm laser irradiation is calculated to be about 44.26% (Figure 7e,f), which is higher than previously reported inorganic photothermal nanomaterials, including Au nanorods (21%), Cu_2_-_X_Se NCs (22%), and Cu_9_S_5_ NCs (25.7%) [12]. Figure 7g shows near-infrared thermal images of a centrifuge tube containing Fe_3_O_4_@SiO_2_@Au/PDA (500 μg/mL) in aqueous solution. As can be seen from the figure, after 7 min laser irradiation, the inside of the centrifuge tube turns red and the temperature rises significantly. These results directly indicate that Fe_3_O_4_@SiO_2_@Au/PDA nanochains exhibit excellent photothermal activity.

Biocompatibility is very important for nanomedicine, since a toxic nanomaterial cannot be applied. Human umbilical vein endothelial cells (HUVEC) are commonly applied to test the cell viability of nanomedicines. In this work, the biocompatibility of the Fe_3_O_4_@SiO_2_@Au/PDA nanochains was tested with an in vitro MTT activity study and apoptosis determination. In vitro biocompatibility and cytotoxicity of Fe_3_O_4_@SiO_2_@Au/PDA nanochains were evaluated using MTT cell assays in human umbilical vein endothelial cells (HUVEC). Under standard conditions (37℃, 5% CO_2_), 10% fetal bovine serum (FBS) was added to DMEM and HUVEC in the logarithmic growth phase was inoculated on 96-well cell culture plates (1×10^4^ cells per well, 200 μL) and incubated for 24 h. Then, fresh DMEM medium containing Fe_3_O_4_@SiO_2_@Au/PDA nanochains (with different gradient concentrations) was replaced and incubated for another 24 h. HUVEC was washed with fresh medium and cell viability was confirmed using a standard MTT assay according to the manufacturer’s protocol. As shown in Figure 8, the Fe_3_O_4_@SiO_2_@Au/PDA nanochains are biocompatible and basically non-toxic. Therefore, this kind of material can be further applied in medicine.

Owing to its wonderful photothermal effect, Fe_3_O_4_@SiO_2_@Au/PDA nanochains can be applied as a PTT reagent. To evaluate the antibacterial properties of Fe_3_O_4_@SiO_2_@Au/PDA nanochains, the inhibition of bacterial growth under different conditions was studied using *Escherichia coli* as a bacterial model. First, three groups of samples with different concentration gradients (0, 200, 400 μg/mL) were prepared and added into LB medium. One group was set as blank control group, and the other two groups were experimental groups with PTT or PTT/Magnetic conditions. Here, PTT means the experiment is conducted by applying an 808 nm NIR laser for 10 min and PTT/Magnetic means both NIR and a magnetic field are applied.

Then, the experimental groups treated under different conditions were placed in an incubator at 37 °C for 12 h. Finally, a certain amount of culture solution was absorbed and coated on the prepared plate, and then placed in an incubator at 37 ℃ for cultivation for 24 h to observe the growth of the colony. Figure 9 shows the antibacterial performance of the Fe_3_O_4_@SiO_2_@Au/PDA nanochains against *Escherichia coli*. Clearly, the *Escherichia coli* in the control group without any external treatment show a good growth trend (the first row of Figure 9), and the increasing concentration of the Fe_3_O_4_@SiO_2_@Au/PDA nanochains shows few effects on the growth of the bacteria, which also agrees that the prepared materials are biocompatible. If no sample was added (the first column of Figure 9), the growth trend of *Escherichia coli* was almost the same, which ruled out the influence of NIR and magnetic field on bacterial growth. Under NIR laser irradiation alone, the Fe_3_O_4_@SiO_2_@Au/PDA nanochains showed a clear sterilization against *Escherichia coli* due to the thermal effect. Moreover, with an increase in the concentration, the sterilization effect was better. This result is because the larger the concentration of the Fe_3_O_4_@SiO_2_@Au/PDA nanochains, the stronger the absorption at 808 nm, and the higher the photothermal effect. As a result, the sterilization effect becomes better.

Under the NIR laser irradiation, if an external magnetic field is introduced, the sterilization effect of Fe_3_O_4_@SiO_2_@Au/PDA nanochains at the same concentration also shows an enhanced effect. During the treatment, *Escherichia coli* sinks to the bottom of the centrifuge tube due to the action of gravity. In the presence of an external magnetic field, the Fe_3_O_4_@SiO_2_@Au/PDA nanochains are subjected to the Lorentz force in the magnetic field and converge towards the magnetic field (Figure 1). At this time, there will be a continuous penetrating force on the bacteria. This may be the reason why the sterilization effect is better than that of simple photothermal sterilization. As a result, the Fe_3_O_4_@SiO_2_@Au/PDA nanochains possess a wonderful coupled magnetolytic–photothermal inhibitory effect (Figure 9). It was found that the Fe_3_O_4_@SiO_2_@Au/PDA nanochains exhibited a unique photothermal effect and thus possess photothermal antibacterial behavior. The magnetic field induced the magnetic nanochains to form a magnetic force which further led to physical damage to the bacteria. As a result, the Fe_3_O_4_@SiO_2_@Au/PDA nanochains possess a coupled magnetic–photothermal antibacterial effect. Here, the magnetic properties of the Fe_3_O_4_@SiO_2_@Au/PDA nanochains do not improve the photothermal property but enhance the antibacterial performance.

Similar to the *Escherichia coli*, the Fe_3_O_4_@SiO_2_@Au/PDA nanochains also show a coupled magnetolytic–photothermal antibacterial performance against *Staphylococcus aureus* (Figure 10). *Staphylococcus aureus* is a Gram-positive bacteria and a common food-borne pathogen. In recent years, there have been numerous reports of food poisoning caused by *Staphylococcus aureus*. Therefore, it is very necessary to study the antibacterial properties of Fe_3_O_4_@SiO_2_@Au/PDA nanochains against *Staphylococcus aureus*. Similar to the results for *Escherichia coli*, when Fe_3_O_4_@SiO_2_@Au/PDA nanochains are co-cultured with *Staphylococcus aureus* without other external conditions, *Staphylococcus aureus* shows a similar growth trend, indicating that the material itself has no killing effect on bacteria. When the Fe_3_O_4_@SiO_2_@Au/PDA nanochain is not added, either treating the solution with 808 nm NIR laser irradiation or a magnetic field, a negative effect on the growth of *Staphylococcus aureus* is observed, indicating that the Fe_3_O_4_@SiO_2_@Au/PDA nanochain is a main reason for the bactericidal effect. In the NIR laser irradiation alone treatment, the antibacterial effect of the 400 μg/mL sample is significantly better than that of the 200 μg/mL sample, because the higher concentration leads to a better photothermal effect. Similarly, the sterilization effect of Fe_3_O_4_@SiO_2_@Au/PDA nanochains against *Staphylococcus aureus* at the same concentration is also enhanced when the NIR laser irradiation and a magnetic field are both applied in comparison to the NIR alone. Obviously, the Fe_3_O_4_@SiO_2_@Au/PDA nanochain produces a magnetic force on the *Staphylococcus aureus* in the presence of an external magnetic field. It was found that this magnetic force is very weak, so the magnetic field alone is not large enough to break the bacteria; thus, the magnetolytic effect cannot be distinguished without applying the NIR irradiation. Under the NIR treatment, the bacteria are inhibited due to the high temperature. Here, the magnetic-field-induced forces improved the antibacterial performance. As a result, the Fe_3_O_4_@SiO_2_@Au/PDA nanochain has a coupled magnetolytic–photothermal antibacterial performance against both *Escherichia coli* and *Staphylococcus aureus* (Figure 1).

## 4. Conclusions

This work reports a core/shell structured Fe_3_O_4_@SiO_2_@Au/PDA nanochains which exhibit a coupled magnetolytic–photothermal antibacterial performance. Using an Fe_3_O_4_ nanochain as the template, the Fe_3_O_4_@SiO_2_@Au/PDA nanochain is prepared with a simple SiO_2_ coating and a subsequent in situ redox-oxidization polymerization of an Au/PDA layer. Owing to the presence of Au and PDA, the Fe_3_O_4_@SiO_2_@Au/PDA nanochain possesses a good photothermal property. The Fe_3_O_4_ also endows the Fe_3_O_4_@SiO_2_@Au/PDA nanochain with soft magnetic characteristics. With the application of NIR irradiation, the Fe_3_O_4_@SiO_2_@Au/PDA nanochain shows excellent antibacterial performance against both *Escherichia coli* and *Staphylococcus aureus* due to the photothermal effect. Interestingly, the magnetic field induces an additional breakage of the bacteria and thus further improves the inhibitory performance. As a result, the Fe_3_O_4_@SiO_2_@Au/PDA nanochain has a coupled magnetolytic–photothermal antibacterial performance against both *Escherichia coli* and *Staphylococcus aureus*. In considering biocompatible surfaces, this kind of material possesses a wide potential for future antibacterial substances and nanomedicine.

## Data Availability

Data is contained within the article.

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
