# Peer review of "Magnetic-Field-Induced Improvement of Photothermal Sterilization Performance by Fe3O4@SiO2@Au/PDA Nanochains"

_materials, 2022, doi:10.3390/ma16010387_

Round 1

Reviewer 1 Report

This is an interesting paper on preparation and antibacterial properties of inorganic-polymer nanocomposites made of Fe3O4/SiO2 nanochains on an Au/PDA support, the so-called nanostrings. The chains are superparamagnetic, and they exhibit auto heating upon photoexcitation under a 808 nm near-infrared laser. As a result, a bactericidal effect is induced against two selected species. The paper is creative, the experiment is well done, employing a variety of methods to prove the concept, and the scope fits nicely with the scope of [Materials]. I believe that this paper can be accepted after minor revision. Please see my comments below.

1.     Page 3, line 106. Please change ‘was add’ for ‘was added’.

2.     Figure 6 is too small. Especially panel (b) cannot be seen without a huge magnification; therefore, it’s difficult to understand the respective discussion on page 9. The same problem holds true for Figure 7, panels (a)-(f). And please remove the second ‘r’ in ‘temperrature’ (Figure 6).

3.     Page 9, line 272. Please change ‘is an amorphous for ‘is amorphous’

4.     Unfortunately, the discussion of superparamagnetic behavior is too terse. Is it possible to explain how the reader should judge whether the particles exhibit superparamagnetic properties or not? Was the hysteresis loop observed at 300 K? Can the graph be enlarged so that the loop becomes visible?

Author Response

Response to referee 1:

Reviewer #1: This is an interesting paper on preparation and antibacterial properties of inorganic-polymer nanocomposites made of Fe3O4/SiO2 nanochains on an Au/PDA support, the so-called nanostrings. The chains are superparamagnetic, and they exhibit auto heating upon photoexcitation under a 808 nm near-infrared laser. As a result, a bactericidal effect is induced against two selected species. The paper is creative, the experiment is well done, employing a variety of methods to prove the concept, and the scope fits nicely with the scope of [Materials]. I believe that this paper can be accepted after minor revision. Please see my comments below. 

Response:Thanks. The relative comments are carefully addressed according to your valuable suggestion.

Question 1. Page 3, line 106. Please change ‘was add’ for ‘was added’.

Response:Thanks. The word was modified in the revision. Page 5, Line 24-25.

Question 2. Figure 6 is too small. Especially panel (b) cannot be seen without a huge magnification; therefore, it’s difficult to understand the respective discussion on page 9. The same problem holds true for Figure 7, panels (a)-(f). And please remove the second ‘r’ in ‘temperrature’ (Figure 6).

Response:Good comments. The Figure 6 and 7 were redrawn and the size of the words was enlarged. The relative word was modified. Thanks for your valuable comment. Page 14, Figure 6 and Page 16, Figure 7.

Question 3. Page 9, line 272. Please change ‘is an amorphous for ‘is amorphous’

Response:Thanks to point such a mistake. The relative words were corrected in the revision. Page 15, Line 5.

Question 4. Unfortunately, the discussion of superparamagnetic behavior is too terse. Is it possible to explain how the reader should judge whether the particles exhibit superparamagnetic properties or not? Was the hysteresis loop observed at 300 K? Can the graph be enlarged so that the loop becomes visible?

Response:Very valuable suggestion. The relative magnetization hysteresis loops were enlarged and the loops were added in the figure 6. The hysteresis loop was measured at 300K. Page 15, Line 18-19.

Reviewer 2 Report

The state of the art does not reflect the connection between the proposed title and the aim of this work. The influence of magnetism on properties (i.e.  photothermal stabilization performance) is not argued. The work contains a complex morpho-structural characterization, but the investigation of magnetic properties and their interpretation in the context of electronic mechanisms that influence the expected effects are almost non-existent. The work can be considered for publication after a complex reorganization, with a clear highlighting of magnetic behavior and its influence on the evolution of properties.

Author Response

Response to referee 2:

Reviewer #2: The state of the art does not reflect the connection between the proposed title and the aim of this work. The influence of magnetism on properties (i.e.  photothermal stabilization performance) is not argued. The work contains a complex morpho-structural characterization, but the investigation of magnetic properties and their interpretation in the context of electronic mechanisms that influence the expected effects are almost non-existent. The work can be considered for publication after a complex reorganization, with a clear highlighting of magnetic behavior and its influence on the evolution of properties.

Response:Thanks for your valuable suggestion. The relative comments are carefully addressed in the revision. The state of the art was modified and the influence of the magnetic field was demonstrated in the scheme 1. Page 8, Scheme 1.

The influence of the magnetic properties on the photothermal stabilization performance is discussed in the revision. In this work, the antibacterial performance of the magnetic composites was investigated. It is found that the Fe3O4@SiO2@Au/PDA nanochains exhibited unique photothermal effect and thus possess the photothermal antibacterials behavior. The magnetic field induce the magnetic nanochains to form a magnetic force which further lead to a physical damage to the bacteria. As a result, the Fe3O4@SiO2@Au/PDA nanochains possess the magnetic-photothermal coupling antibacterial effect. Here, the magnetic properties of the Fe3O4@SiO2@Au/PDA nanochains do not improve the photothermal property but enhance the antibacterial performance. Page 19, Line 26-30.

Reviewer 3 Report

The research about antibacterial methods that do not cause resistance led the authors to develop a Fe3O4@SiO2@Au/PDA nanochain capable of enhancing photothermal antibacterial properties through electromagnetic field induction, which generates new perspectives in clinical applications. As the paper presents some gaps, some suggestions must be taken into consideration in order to make it better as well as more complete.

The authors started the contextualization based on treatments for "diseases caused by bacterial infections" (line 32), but this concept is very broad when applied to systemic infections. 

But in the case of local treatment such as skin lesions or some cancers, the treatment can be promising, once the application would have precise targeting, avoiding greater risks to the patient. It would be interesting to specify some of these clinical applications so that the reader can understand exactly what is being proposed.

The article states that the antibacterial activity of the Fe3O4@SiO2@Au/PDA nanocomposite is due to "thermal ablation" (line 41) caused by the photothermal properties induced by the electromagnetic field, which leads to bacteria lysis. Photoactivation can generate reactive oxygen species, which, in excess, can cause damage to adjacent tissues. Specific tests for ROS would enrich the results.

The choice of the elements used for the synthesis of the nanochain was very well described. However, the choice of polydopamine could have been better explained in the text.

The methodology was well-defined, but it does not specify the electromagnetic field model used. How was it applied? It is necessary to give details about this method.

Line 37 – 40 During the second paragraph the author explains the advantages of the PTT as a new technology for the treatment of bacterial infections. It would be interesting to add some examples of disadvantages too.

Line 78 – 80 What type of infection is the author referring to when he says: “with 500 μg/mL, the temperature rises to 57 ℃ if irradiated by 808 nm laser”? Does it not harm the cells around with a local or systemic infection?

Overall, the figure legends need more information and details.

Figure 4: The scale needs to be reviewed and the legend to be improved with more information.

Line 219 The authors affirmed “The introduction of SiO2 coating is to make the chain structure more stable.” What assay demonstrated this with the property? I suggest evaluating the Z-potential and Dynamic Light Scattering of all nanoparticles. One important point that needs to be evaluated is the behavior of the nanoparticles in different pH, once some bacteria alter this parameter in the environment.

Line 238 the correct identification of the Figures needs to be reviewed

Figure 6C: I would appreciate if the chemical groups were identified.

Figure 7 A and D need to amplify the letters (the size of the letters in A and D need to be bigger?)

Figure 8:  There is a complete lack of method description about cell viability with HUVEC cells in the Experimental section. On the cell viability test, HUVECs cells were used. However, why this type of cell was chosen is not explained/ discussed. This topic may be improved with a better discussion. In addition, it is not informed if the NIR laser or magnetic field has effects on cell growth (line 349-361). It is necessary to present the positive control in this assay. Evaluating high concentrations (400ug, 500ug, 600ug, etc) is crucial to evaluate the necessary concentration to promote cell death.

Line 350 - The “apoptosis determination” is not demonstrated with only MTT assay. I suggest another assay in order to quantify/evaluate apoptosis, such as Annexin-V or Caspase assays.  

Figure 9: The concentration of 400ug is able to inhibit bacterial growth, but it was not evaluated in HUVEC cells. 

It is important to evaluate the effect in HUVEC cells using the same treatment in bacteria ( PTT and PTT+magnetic), it will demonstrate specificity in this innovative treatment.

Line 384 the correct identification of the Figures needs to be reviewed.

In the antibacterial tests with gram-negative and gram-positive bacteria, they could have performed specific assays using Minimum Inhibitory and Bactericidal Concentration (MIC and WBC), IC50 (concentration that inhibits 50% growth) or measurement of the zone of inhibition to quantify the data and generate robust data in graphs.

Line 418-420: There is misinformation about the concentrations here “In the sole NIR laser irradiation, the sterilization of 400 ug/mL sample is significantly better than that of 200 ug/mL sample because the higher the concentration, the better the photothermal effect”. It is vital to improve the discussion about it.

Author Response

Response to referee 3:

Reviewer 3: The research about antibacterial methods that do not cause resistance led the authors to develop a Fe3O4@SiO2@Au/PDA nanochain capable of enhancing photothermal antibacterial properties through electromagnetic field induction, which generates new perspectives in clinical applications. As the paper presents some gaps, some suggestions must be taken into consideration in order to make it better as well as more complete.

Answer: Thanks to the comment, the relative modification was added in the revision.

Question 1: The authors started the contextualization based on treatments for "diseases caused by bacterial infections" (line 32), but this concept is very broad when applied to systemic infections. But in the case of local treatment such as skin lesions or some cancers, the treatment can be promising, once the application would have precise targeting, avoiding greater risks to the patient. It would be interesting to specify some of these clinical applications so that the reader can understand exactly what is being proposed.

Answer: Good comment. In spired by this valuable suggestion, the local treatment skin lesions and breast cancer are mentioned in the revision.

Question 2: The article states that the antibacterial activity of the Fe3O4@SiO2@Au/PDA nanocomposite is due to "thermal ablation" (line 41) caused by the photothermal properties induced by the electromagnetic field, which leads to bacteria lysis. Photoactivation can generate reactive oxygen species, which, in excess, can cause damage to adjacent tissues. Specific tests for ROS would enrich the results.

Answer: This is a very good comment. It is widely reported that the photoactivation can generate reactive oxygen species, which, in excess, can cause damage to adjacent tissues. In our most recently results, we also found that the nanocomposite can catalyze the oxidation of TMB by H2O2, which also demonstrates the generation of reactive oxygen species. It is a very good suggestion to conduct the specific tests for ROS since it can enrich the result. Unfortunately, it is still difficult for us to confirm this point since the serious COVID-19 in all over the China very recently. Moreover, in this work, our target is to find the photothermal effect of the nanochains on the bacterialysis, the specific tests for ROS and the effects of the ROS on the antibacterial will be carefully discussed and decoupled in the future.

Question 3: The choice of the elements used for the synthesis of the nanochain was very well described. However, the choice of polydopamine could have been better explained in the text.

Answer: Good suggestion. The choice of the polydopamine has been added in the text.

Question 4: The methodology was well-defined, but it does not specify the electromagnetic field model used. How was it applied? It is necessary to give details about this method.

Answer: Good suggestion. The magnetic field applying method was added in the manuscript. Bacterial suspensions were treated with NIR laser (L, 2.5 W cm-2) and rotating magnetic field (RMF, 15 mT, 350 rpm), and the three groups were labeled as Control, PTT, and PTT + Magnet, respectively. Except for the control group, the treatment time was 6 min, where PTT + Magnet was the combined action of NIR laser and rotating magnetic field for 6 min. During the magnetic treatment, the magnetic field was conducted by using a magnetic stir which directly performed under the solution.

Question 5: Line 37 – 40 During the second paragraph the author explains the advantages of the PTT as a new technology for the treatment of bacterial infections. It would be interesting to add some examples of disadvantages too.

Answer: Yes, some disadvantages such as the non-specifity and low penetration were added in the manuscript. “In considering that the PTT also shows some drawbacks such as the non-specifity and low penetration, intensity works have been conducted to improve the PTT performance and clarify the PTT mechanism.”

Question 6: Line 78 – 80 What type of infection is the author referring to when he says: “with 500 μg/mL, the temperature rises to 57 ℃ if irradiated by 808 nm laser”? Does it not harm the cells around with a local or systemic infection?

Answer: Yes, this kind of temperature is high for breaking the bacteria. Actually, this test is just to demonstrate that the present nanomaterials exhibit wonderful PTT effect and the temperature can be increased to 57 ℃ if irradiated by 808 nm laser with 500 μg/mL. During the actually application, the dosage of the nanomaterials and the NIR irradiation will be optimized and the temperature will be tuned to be the best one. It is indeed harm the cells with a local or systemic infection under such a high temperature. However, the optimization will be conducted before the nanomaterials being applied to treat cell and body.

Question 7: Overall, the figure legends need more information and details.

Figure 4: The scale needs to be reviewed and the legend to be improved with more information.

Answer: Good comment. The scale and legend of Figure 4 are checked in the revision.

Question 8: Line 219 The authors affirmed “The introduction of SiO2 coating is to make the chain structure more stable.” What assay demonstrated this with the property? I suggest evaluating the Z-potential and Dynamic Light Scattering of all nanoparticles. One important point that needs to be evaluated is the behavior of the nanoparticles in different pH, once some bacteria alter this parameter in the environment.

Answer: This is a very good comment. Our experimental results indicate that the Fe3O4 chains are easily to be destroyed during the terrible sonication etc. However, after the SiO2 coating, all the Fe3O4 chains are encapsulated by the tough shell and thus the chain structure is more stable. The Z-potential and Dynamic Light Scattering of all nanoparticles are very important and these information can characterize the state of the microparticles. Unfortunately, the nanochains are very easily to be aggregated during the test and the collecting data cannot give any valuable information. It is very sorry that our students all left the lab very recently due to the serious COVID-19 in all over the China very recently. The present particles are stable in different pH. However, if the pH is high, the particles can be etched due to the dissolution of SiO2 coating.

Question 9: Line 238 the correct identification of the Figures needs to be reviewed

Answer: Thanks for your comment. The caption of the figure 5 is corrected in the revision.

Question 10: Figure 6C: I would appreciate if the chemical groups were identified.

Answer: Good comment. The diffraction peaks in the XRD are indexed and the related description is added in the revision.

Question 11: Figure 7 A and D need to amplify the letters (the size of the letters in A and D need to be bigger?)

Answer: Good comment. The size of the letters in A and D are amplified to be bigger and the relative images are modified in the revision.

Question 12: Figure 8:  There is a complete lack of method description about cell viability with HUVEC cells in the Experimental section. On the cell viability test, HUVECs cells were used. However, why this type of cell was chosen is not explained/ discussed. This topic may be improved with a better discussion. In addition, it is not informed if the NIR laser or magnetic field has effects on cell growth (line 349-361). It is necessary to present the positive control in this assay. Evaluating high concentrations (400ug, 500ug, 600ug, etc) is crucial to evaluate the necessary concentration to promote cell death.

Answer: Good comment. The method description about cell viability with HUVEC cells was added main text as shown below the figure 8. Here, this work is focused on the influence of the nanocomposites on the antibacterial performance, thus the cell viability test is just simply discussed biocompatibility of the nanocomposite. The cell viability with HUVEC cells was also added into the experiment section in the revision. Moreover, the influence of the NIR laser or magnetic field on cell growth has not been conducted, since the nanocomposites are too large to enter the small cells. Our future work is focused on fabrication small anisotropic nanocomposites which can enter the cell and lead to the death of the cell under the NIR laser or magnetic field. This work is on going very recently and the results will be published elsewhere. In this work, the cell death is not our concentrated topic and the relative experiments have not been well clarified. Inspired by your valuable suggestions, the influence of the NIR laser, magnetic field, and the high concentrations will be done to well demonstrate the coupling behavior.

Question 13: Line 350 - The “apoptosis determination” is not demonstrated with only MTT assay. I suggest another assay in order to quantify/evaluate apoptosis, such as Annexin-V or Caspase assays.

Answer: Good comment. Annexin-V or Caspase assays are very good to quantify/evaluate apoptosis. Unfortunately, the students all left the lab very recently due to the serious COVID-19 in all over the China very recently. It is very difficult for us to conduct such experiments. In this work, the MTT assay is just to confirm the biocompatibility of the Fe3O4@SiO2@Au/PDA nanochains, the Annexin-V or Caspase assays will be further examined in our future work.

Question 14: Figure 9: The concentration of 400ug is able to inhibit bacterial growth, but it was not evaluated in HUVEC cells. It is important to evaluate the effect in HUVEC cells using the same treatment in bacteria ( PTT and PTT+magnetic), it will demonstrate specificity in this innovative treatment.

Answer: Good comment. Our examination found that the higher concentration is harmful for the cell growth. Due to the experimental condition, the high concentration has not been conducted. In this work, the MTT assay is just to confirm the biocompatibility of the Fe3O4@SiO2@Au/PDA nanochains, thus the low concentrations were tested. Moreover, the size of the Fe3O4@SiO2@Au/PDA nanochains is large, which is not suitable for evaluate the effect in HUVEC cells using the same treatment in bacteria (PTT and PTT+magnetic). Inspired by this valuable suggestion, our future work will be focused on fabrication small anisotropic nanocomposites which can enter the cell and lead to the death of the cell under the NIR laser or magnetic field.

Question 15: Line 384 the correct identification of the Figures needs to be reviewed.

In the antibacterial tests with gram-negative and gram-positive bacteria, they could have performed specific assays using Minimum Inhibitory and Bactericidal Concentration (MIC and WBC), IC50 (concentration that inhibits 50% growth) or measurement of the zone of inhibition to quantify the data and generate robust data in graphs.

Answer: Thanks, the identification of the Figures was corrected in the revision. It is very right that the Minimum Inhibitory and Bactericidal Concentration (MIC and WBC), IC50 (concentration that inhibits 50% growth) or measurement of the zone of inhibition to quantify the data and generate robust data are important for evaluating the antibacterial performance. In this work, this novel magnetic field improving antibacterial area was just started and more in-depth research should be conducted to clarify the magnetic field induced mechanism. In spired by this valuable comment, the MIC, WBC, IC50 tests will be discussed and published elsewhere.

Question 16: Line 418-420: There is misinformation about the concentrations here “In the sole NIR laser irradiation, the sterilization of 400 ug/mL sample is significantly better than that of 200 ug/mL sample because the higher the concentration, the better the photothermal effect”. It is vital to improve the discussion about it.

Answer: Thanks, the relative description was modified in the revision.

Round 2

Reviewer 2 Report

The manuscript has been considerably improved according to the ideas proposed. In my opinion, this new version can be considered for publication.

Author Response

Thanks for your approving of publication

Reviewer 3 Report

 I am satisfied with the corrected version submitted and the responses.